# *Ulva lactuca* Extract and Fractions as Seed Priming Agents Mitigate Salinity Stress in Tomato Seedlings

**DOI:** 10.3390/plants10061104

**Published:** 2021-05-30

**Authors:** Mohammed El Mehdi El Boukhari, Mustapha Barakate, Nadia Choumani, Youness Bouhia, Karim Lyamlouli

**Affiliations:** 1Biodiversity and Plant Sciences Program, Mohammed 6 Polytechnic University (UM6P), AgroBioScience, Benguerir 43150, Morocco; mohamed.elboukhari@um6p.ma (M.E.M.E.B.); Mustapha.BARAKATE@um6p.ma (M.B.); Youness.BOUHIA@um6p.ma (Y.B.); 2Laboratory of Microbial Biotechnology, AgroSciences and Environment, Faculty of Sciences Semlalia, Cadi Ayyad University, Marrakesh 40000, Morocco; 3Department of Chemical and Biochemical Sciences-Green Process Engineering (CBS-GPE), Mohammed VI Polytechnic University (UM6P), Benguerir 43150, Morocco; nadia.choumani@um6p.ma

**Keywords:** seaweed extract, *Ulva lactuca*, solvent–solvent fractionation, abiotic stress, plant biostimulant

## Abstract

The present study investigates the effect of *Ulva lactuca* extract as seed-priming agent for tomato plants under optimal and salinity stress conditions. The aims of this experiment were to assess the effect of seed priming using *Ulva lactuca* extract in alleviating the salinity stress tomato plants were subjected to, and to find out the possible mechanism of actions behind such a positive effect via means of fractionation of the crude extract and characterization. Salinity application decreased the plant biomass and altered different physiological traits of tomato. However, the application of *Ulva lactuca* methanol extract (ME) and its fractions (residual fraction (RF), chloroform fraction (CF), butanol fraction (BF), and hexane fraction (HF)) at 1 mg·mL^−1^ as seed priming substances attenuated the negative effects of salinity on tomato seedlings. Under salinity stress conditions, RF application increased the tomato fresh weight; while ME, RF, and HF treatments significantly decreased the hydrogen peroxide (H_2_O_2_) concentration and antioxidant activity in tomato plants. The biochemical analyses of *Ulva lactuca* extract and fractions showed that the RF recorded the highest concentration of glycine betaine, while the ME was the part with the highest concentrations of total phenols and soluble sugars. This suggests that these compounds might play a key role in the mechanism by which seaweed extracts mitigate salinity stress on plants.

## 1. Introduction

One of the big challenges the humankind is facing today is to ensure that there is enough food for an increasingly growing population. In 2019, practically 690 million people all over the world went hungry [1]. Enhancing food production in a sustainable way is a key element to reduce the hunger in the world [2]. However, achieving this goal is connected with many challenges, such as nutrient imbalance, water scarcity, and degradation of arable lands [3,4]. The latter challenge is the result of many processes that occur either naturally or due to human activity. Salinity is a clear example of such processes. Shahid et al. [5] reported that 10% of the arable lands are affected by sodicity and salinity, while 25% to 30% of the irrigated land area are suffering from salinity and consequently become commercially uncompetitive.

Salinity alters many mechanisms playing a key role in plant growth and development, such as decreasing its shoot and root biomass, increasing the production of reactive oxygen species (ROS), increasing the biosynthesis of antioxidant compounds, and decreasing its photosynthetic activity [6,7]. Different approaches were proposed to attenuate the negative impact of salinity on crops. Irrigation and nutrient management, crop rotation, reduced tillage, mulching, cropping tolerant crops, and phytoremediation are instances of such strategies [8]. Using plant biostimulants (PB) is another efficient solution that was proven to play a key role in minimizing the deleterious effects of salinity [9,10,11]. Seaweed extracts (SE) are currently regarded as in important PB category with beneficial effects on alleviating abiotic stresses on cropped plants such as salinity [12,13]. Those effects are mainly related to their ability to attenuate the oxidative damage on stressed plants by scavenging ROS, regulating their antioxidant activity through the enhancement of total phenols and enzymatic and non-enzymatic antioxidants, reducing ionic imbalances, improving electrical leakage, reducing lipid peroxidation, and enhancing the accumulation of osmo-protectants [14,15,16,17,18]. Moreover, such positive effects are often associated with an improvement of the photosynthetic activity, sugars accumulation, and nutritional status leading to an improvement of plant biomass and quality [19,20,21].

SE biostimulants are either directly applied in soil as a foliar spay, or as a seed priming agent [22]. The latter application would be an effective route to attenuate salinity stress menacing plants during germination and early growth stages [23]. For instance, the application of different SE as seed priming agents on *Vigna sinensis* and *Zea mays* enhanced the germination rate, dry biomass, carbohydrates content, protease and amylase activities of plants during the seedling stage [24].

However, despite several research investigations, the mechanisms of action by which SE biostimulants affect the plant stress tolerance are still unclear. Consequently, the replicability of such positive effects on plant growth and development after SE application is questionable. Furthermore, the last European union fertilizing product regulation insists on this point and pushes SE manufacturers to respect the claims by pinpointing their origin on product labels [13]. Thus, the development of SE biostimulants is becoming a challenge, knowing that the composition of SE biostimulants is highly complex, hence predicting the occurring synergies and antagonisms between the bioactive molecules is intricate. Moreover, studies investigating the effect of those products rarely consider the specific response of each growing stage of the plant, which is of paramount importance as to have a complete picture with respect to physiological changes.

In this context, this study was proposed to understand the plausible mechanism of actions by which SE mitigate salinity stress on tomato seedlings. To do so, *Ulva lactuca* crude extract was fractionated throughout a solvent–solvent extraction using a cascade of solvents with ascending polarity in order to identify groups of molecules present in each fraction. Then, the crude extract and fractions were applied separately on tomato seeds as priming agents. Consequently, correlating positive traits recorded on tomato seedling growth and physiology with the biochemical content of each fraction would give some routes in the way of deciphering the mechanism of action by which the *Ulva lactuca* extract may plausibly act.

## 2. Results

### 2.1. Biochemical Characterization of Ulva lactuca Extract Fractions and Yield

The biochemical analyses of methanol extract (ME) of *Ulva lactuca* and its fractions showed that the total phenols were significantly higher in ME (112.64 mg Gallic acid equivalent (GAE)·g^−1^ dry weight DW) in comparison with other fractions. Similarly, the total flavonoids and soluble sugars were significantly higher in the ME (4.87 mg Quercitine·g^−1^ DW) and (0.296 mg Glucose·g^−1^ DW) for flavonoids and sugars, respectively. However, the glycine betaine concentration was significantly higher in the residual fraction (RF) with 8.033 mg·g^−1^ DW (Table 1).

### 2.2. Effect of Ulva lactuca Extract on Tomato Growth Traits

The shoot fresh weight of tomato plants had significantly decreased when the EC levels were increased (*p* < 0.05). However, priming tomato seeds with ME of *Ulva lactuca* enhanced significantly the shoot fresh weight of plants irrigated with 4 dS·m^−1^ nutrient solution in comparison with the control by 39.88%. Furthermore, RF treatment gave plants irrigated with 8 dS·m^−1^ nutrient solution a highly significant advantage over the control by enhancing their fresh weight by 70.49%. The application of *Ulva lactuca* ME and its fractions did not induce any significant enhancement of the leaf area, root length, and root projected area (Table 2).

### 2.3. Effect of Ulva lactuca Extract on Tomato Antioxidant Activity

Application of salinity treatments (4 and 8 dS·m^−1^) on tomato plants resulted in an increase of their antioxidant activity (*p* < 0.001) (Figure 1). Priming tomato seeds with *Ulva lactuca* ME and its fractions did not induce any significant differences with non-treated plants when they were irrigated with 2 dS·m^−1^ nutrient solution. However, ME and RF applications induced a highly significant decrease of antioxidant activity in plants supplied with 4 dS·m^−1^ nutrient solution (22.9%) in comparison with the control. Similarly, ME and HF deceased significantly the antioxidant activity of tomato plants by 23.6% and 25.8%, respectively, when compared to the control under 8 dS·m^−1^ stress conditions. Surprisingly, BF application enhanced significantly the tomato antioxidant activity in comparison with the control at 4 dS·m^−1^.

### 2.4. Effect of Ulva lactuca Extract Fractions on Tomato Hydrogen Peroxide Concentration

The differences of H_2_O_2_ concentrations in tomato leaves were highly significant between different salinity treatments (*p* < 0.001) (Figure 2). For instance, 484.96 nmol·g^−1^ fresh weight (FW) was recorded in control plants growing in optimal conditions of 2 dS·m^−1^. While in 8 dS·m^−1^ EC growth conditions, the H_2_O_2_ concentration increased to 661.33 nmol·g^−1^ FW. The application of *Ulva lactuca* ME and its fractions decreased significantly the H_2_O_2_ concentration of tomato leaves, particularly the ME which lowered this concentration by 33%, 37%, and 21.02% at 2, 4, and 8 dS·m^−1^, respectively, in comparison with the control.

### 2.5. Effect of Ulva lactuca Extract Fractions on Tomato Soluble Sugars and Total Proteins Content

Highly significant differences between soluble sugars (*p* < 0.01) (Figure 3) and the total protein content (*p* < 0.001) (Figure 4) of tomato leaves grown in different salinity levels were recorded. For example, soluble sugars and the total protein content were, respectively, 3.88 and 0.86 mg·g^−1^ FW for plants treated with the RF at 2 dS·m^−1^ growth conditions. These concentrations dropped to 3.24 and 0.59 mg·g^−1^ FW for soluble sugars and the total protein content, respectively, at 4 dS·m^−1^. However, the RF application enhanced significantly the concentration of soluble sugars of tomato leaves when compared to the control at 4 dS·m^−1^ salinity level. Moreover, both ME and RF applications enhanced significantly the total proteins content of plants supplied with 8 dS·m^−1^ nutrient solution in comparison with the control by 191% and 163%, respectively.

### 2.6. Effect of Ulva lactuca Extract Fractions on Tomato Leaves Chlorophyll and Carotenoides

The salinity application reduced significantly the total chlorophyll content (chlorophyll *a* + *b*) of tomato leaves (*p* < 0.001). Yet, the application of the RF enhanced significantly the total chlorophyll content of treated plants in comparison with the control by 63.84%, 67% and 86.81% at 2, 4, and 8 dS·m^−1^, respectively. Moreover, HF and ME applications enhanced significantly the total chlorophyll content of tomato leaves when compared to the control at 8 dS·m^−1^ salinity level (Figure 5).

Differences of the total carotenoid content of tomato leaves between plants supplied by 2 dS·m^−1^ and those supplied by 8 dS·m^−1^ nutrient solutions were significant (*p* < 0.05). However, no significant differences between SE treatments and the control were recorded (Figure 6).

### 2.7. Principal Component Analysis (PCA)

PCA was performed to understand different correlations between the studied parameters recorded on tomato plants (antioxidant activity, hydrogen peroxide concentration, total chlorophyll, soluble sugars, total proteins, total carotenoids, leaf area, fresh biomass, root projected area, root length, root volume, and root average diameter) at different salinity levels (Figure 7). The first component described 37.2%, 37.6%, and 44.1% of the variables at 2, 4, and 8 dS·m^−1^ salinity levels, respectively. The second component described 23.4%, 22.7%, and 18.3% of the variables at 2, 4, and 8 dS·m^−1^ salinity levels, respectively. At optimal growth conditions (2 dS·m^−1^), the first component was more correlated with the total chlorophyll content, leaf area, and antioxidant activity. The second component was more correlated with the root projected area, root length, and root volume (Figure 7A). At mild salinity conditions (4 dS·m^−1^), the first component was closely associated with the fresh biomass and antioxidant activity. While the second component was more correlated to soluble sugars (Figure 7B). At high salinity conditions (8 dS·m^−1^), the first component was more correlated with the total chlorophyll, antioxidant activity, and root length. While the second component was more correlated with the H_2_O_2_ concentration and total carotenoids (Figure 7C).

## 3. Discussion

Solvent–solvent fractionation is an approach used to separate a crude extract into different groups of fractions having a distinct criterion in common (molecular weight, solubility, etc.) by a confrontation of the extract with different solvents. This tool is used among other fractionation techniques, e.g., chromatography with the objective to reduce the complexity of a given extract mixture as to determine significant correlation between active molecules and the resulting effect on a biological agent [25]. In our study, the solvent–solvent fractionation of *Ulva lactuca* extract prior to biochemical characterization enabled the determination of molecules or/and groups of molecules that might play a key role in the mechanism of action by which the aforementioned extract attenuates the salinity stress in tomato at the seedling stage. The biochemical analyses of *Ulva lactuca* extract showed that the ME recorded the highest values of total phenols and total flavonoids in comparison with other fractions, while the RF showed the highest concentration of glycine betaine. These findings agree with those of Bannour et al. [26] who found that the methanol extract of *Calligonum azel* obtained through Soxhlet and pressurized liquid extractions had the highest concentration of total phenolic compounds when compared to hexane and water extracts.

Most of studies dealing with seed priming were mainly focused on the germination stage. Moreover, the effect of SE on plants would be different from one stage to another. Consequently, our study focused on the seedling tomato stage to understand to some extent the effect of the *Ulva lactuca* extract as a priming agent on tomato early vegetative growth.

Salinity application reduced significantly shoot and root biomass of tomato plants, decreased soluble sugars, total protein content, and chlorophyll concentration of tomato leaves, and increased their antioxidant activity and production of hydrogen peroxide. However, priming tomato seeds with *Ulva lactuca* ME and RF succeeded in enhancing shoot weight of tomato seedlings growing in 4 dS·m^−1^ and 8 dS·m^−1^ growth conditions, respectively. Similar effects were reported by Chanthini et al. [27] who found that priming cherry tomato seeds with an *Ulva flexuosa* SE promoted seedling shoot and root biomass. This effect could be attributed to the high concentration of glycine betaine in the RF and ME in comparison with other fractions. This could be supported by the findings of Cisse et al. [28], which revealed that a glycine betaine exogenous application enhanced significantly biomass traits, e.g., the plant height and number of leaves of *Dalbergia odorifera* at the seedling stage under mild and severe salinity growth conditions. This enhancement was associated with an increase in the water use efficiency, photosynthetic activity, transpiration, phenolic compounds and ROS scavenging activity. Osmotic adjustment, ROS scavenging and subcellular structures stabilization are instances of mechanisms by which glycine betaine is involved in mitigating the salt stress in plants [29]. Glycine betaine synthesis in plants consists in converting choline into betaine aldehyde via the choline monooxygenase (CMO) enzyme and, thereafter, betaine aldehyde dehydrogenase (BADH), a NAD^+^ dependent enzyme, produces glycine betaine. These enzymes are basically present in the chloroplast stroma and their activity is enhanced when salt stress occurs [30]. However, even though the final yield of tomato fruits was significantly enhanced, osmo-priming tomato seeds with two commercial *Ascophyllum nodosum* SE reduced significantly the fresh weight of tomato plants [31]. This reduction would be the result of the presence of high concentrations of sorbitol and mannitol in the used extracts, as stated by the authors.

Salinity stress results in an oxidative stress leading to overproduction of ROS [32]. These are the product of activation or reduction of O_2_ inducing the formation of singlet oxygen (^1^O_2_), superoxide (O_2_^−^), hydroxyl radical (HO^−^), and hydrogen peroxide (H_2_O_2_) [33]. ROS are thought to trigger dysfunctions in proteins, lipids, and deoxyribonucleic acid (DNA) [34]. The application of *Ulva lactuca* extract enabled reduction of ROS production in tomato plants. ME decreased H_2_O_2_ production in all growth conditions, while RF, BF, and HF succeeded in reducing this concentration in high salinity levels growth conditions only. This reduction could be attributed to the presence of antioxidants in these extracts, like total phenols which were higher in the ME in comparison with other fractions. Our finding are supported by the works of Patel et al. [15] who linked the reduced H_2_O_2_ production in *Triticum durum* plants subjected to salinity and drought stresses and treated with *Kappaphycus alvarezii* sap to the increased concentration of non-enzymatic antioxidants such as total phenols and to the enhanced expression of superoxide dismutase (SOD) and catalase (CAT) genes.

When plants are encountering an abiotic stress, their antioxidant activity is tending to increase in response to the higher production of ROS [35,36]. PCA analyses confirm this statement as the increase of antioxidant activity at optimal and salinity stress conditions was associated with the reduction of growth and physiological traits of tomato plants like the total chlorophyll content and fresh biomass. The application of *Ulva lactuca* ME, HF, and RF reduced the antioxidant activity of stressed tomatoes when compared with the control. This effect witnesses the ability of these extracts to attenuate the adverse effects of salinity on tomato. These results are in line with those reported by Dawange and Jaiswar [37] who noticed a decrease in the free radical scavenging activity of *Gracilaria corticata* seaweed-treated by an *Ascophyllum nodosum* SE. In an attempt to decipher the mechanism of action behind the decrease of antioxidant activity of asparagus plants subjected to salinity, Al-Ghamdi and Elansary [38] spotted an up-regulation of the redox responsive genes of GPX3 and APX1 in treated plants with *Ascophyllum nodosum* SE in comparison with the non-treated ones. However, Di Mola et al. [39] stated that an application of a tropical plant extract biostimulant on baby rocket plants under nitrogen deficiency conditions lead to an increase of the antioxidant activity of leaves. Similar findings were reported by Jang et al. [40] who reported an increase of *Peucedanum japonicum* antioxidant activity after application of soybean, Chinese chive, onion, and tomato extracts.

The ME and HF of *Ulva lactuca* were found to enhance the total protein content of tomato leaves, which is in agreement with the findings of González-González et al. [41] who reported an increase of protein accumulation in tomato plants treated with *Padina gymnospora* SE. As a possible scenario for this enhancement, these authors hypothesized that plants treated with biostimulants might have the ability to absorb more of the essential elements. Moreover, in our research, the application of *Ulva lactuca* extract induced a slight increase of soluble sugars in stressed tomato leaves. Similarly, a rich fraction on carbohydrates of an *Ascophyllum nodosum* extract was found to decrease the reduction of soluble sugars of heat stressed tomatoes both in leaf and flower tissues [42].

Several studies in the literature show evidence of seaweed extracts enhancing the chlorophyll content of treated plants like mung bean [43], garden cress and wheat [44]. The enhancement of plants’ chlorophyll content following an application of seaweed extracts could be attributed to the presence of betaines in these extracts [45]. Moreover, Hamani et al. [46] showed that an application of 5 mM glycine betaine enhanced the chlorophyll content of cotton under salinity stress. Furthermore, they pinpointed that the enhancement of the total chlorophyll content was positively correlated with net photosynthetic rate, which might be one of the factors that was responsible for the salinity stress mitigation. Our study confirms this statement as plants treated with the RF of *Ulva lactuca* extract, having the highest concentration of glycine betaine, recorded the highest chlorophyll content in all growth conditions (2, 4 and 8 dS·m^−1^). However, an application of two commercial SE based on *Ascophyllum nodosum* and *Ecklonia maxima* on potato did not alter its leave chlorophyll content [47]. Interestingly, they found that chlorophyll content variations were related mainly to cultivar and cropping season factors.

## 4. Material and Methods

### 4.1. Ulva lactuca Sampling and Characterization

*Ulva lactuca* seaweeds were harvested from El Jadida city of Morocco (33°14′38.5″ N 8°32′36.8″ W) in August 2019. After being washed thoroughly by tap water to eliminate sand and epiphytes, the seaweeds were oven dried at 60 °C for 72 h, ground and sieved to obtain particles less than 1 mm of diameter.

### 4.2. Solvent Solvent Extraction and Fractionation

The amount of 10 g of dry seaweed was extracted in a Soxhlet apparatus using methanol 70% as a solvent in a ratio of 1/10 (*w/v*) for 3 h and at a temperature of 40 °C. Afterwards, different fractions of the methanol extract (ME) were performed through a solvent–solvent extraction in a cascade of solvents with ascending polarity. The hexane fraction (HF) was obtained by extracting the ME in hexane using a ME phase:hexane volume ratio 1:1 (*v/v*). Extraction was repeated thrice, and the phases were combined. Similarly, the chloroform fraction (CF)and the butanol fraction (BF) were obtained by extracting the organic phase (ME) in chloroform and n-butanol consecutively. The residual phase was considered as the residual fraction (RF). ME and its fractions were dried in a rotavapor and extraction yield was recorded as % *w/w* (dry extract: seaweed).

### 4.3. Seaweed Extract Fractions Characterization

#### 4.3.1. Determination of Total Phenolic Content and Total Flavonoids

Samples were prepared by dissolving 5 mg of dry extracts in 5 mL of water. The total phenolic content was determined according to the protocol described by Ainsworth and Gillespie [48]. A volume of 230 µL of 10% (*v*/*v*) folin ciocalteu reagent was added to 100 µL of each sample or blank. Then, 800 µL of 700 mM Na_2_CO_3_ was added to the mixture. After incubating for 2 h, absorbance was determined at 765 nm. A standard curve was prepared using solutions with different concentrations of Gallic acid (0, 50, 100, 150, 250, and 500 µg·mL^−1^). The total phenolic content was expressed as mg GAE·g^−1^ DW.

The total flavonoids were estimated based on the method described by Herald et al. [49] with some modifications. Briefly, 1 mL of extract was added to 4 mL of deionized water and 0.3 mL of 5% NaNO_2_ and vortexed for 5 min. Then 0.3 mL of 10% AlCl_3_ was added to the mixture and was allowed to stand for 6 min. Finally, 2 mL of 1 M NaOH was added and total volume was adjusted to 10 mL. Absorbance was determined at 510 nm. The standard curve was prepared using solutions having different concentrations of quercetin (0, 20, 40, 60, 80, 100 mg·L^−1^). The total flavonoid concentration was expressed as mg quercetin·g^−1^ DW.

#### 4.3.2. Determination of Glycine Betaine Concentration

Glycine betaine was determined as described by Grieve and Grattan [50]. Extracts were diluted 1:1 with 2 N H_2_SO_4_. Aliquots of 0.5 mL were cooled in ice water for 1 h. A volume of 0.2 mL of cold KI-I_2_ was added and the mixture was stirred. Samples were stored for 16 h at 0–4 °C and then centrifuged at 10,000× *g* for 15 min at 0 °C. After removing the supernatant, the periodide complex was dissolved in 9 mL of 1,2-dichloroethane. Finally, the absorbance was measured at 365 nm after 2 h. Standard solutions of glycine betaine (50–200 µg·mL^−1^) were prepared in 1 N H_2_SO_4_.

#### 4.3.3. Quantification of Soluble Sugars

Soluble sugars were estimated based on the protocol described by Dubois et al. [51]. The amount of 2 mL of a sample was added to 0.05 mL of phenol 80% (*w/w*). Next, 5 mL of sulfuric acid 95.5% was added rapidly to the solution. The homogenate was allowed to stand 10 min before shaking and placing in a water bath at 25 °C. Absorbance was read at 490 nm. Standard curve was prepared using sugar solutions containing from 10 to 70 µg of glucose. Soluble sugars were given as mg glucose g^−1^ DW.

### 4.4. Mode of Application of Seaweed Extracts as Seed Priming Agents

Tomato seeds were surface sterilized using sodium hypochlorite 2% for 5 min before being rinsed three times with sterile deionized water. Thereafter, the seeds were soaked in different extract fractions (MF, HF, CF, BF, and RF) or water (control plants) at a temperature of 25 ± 1 °C under gentle shaking in a concentration of 1 mg·mL^−1^ (dry extract: water (*w/v*)). Dry extracts were resuspended in distilled water. A concentration of 1 mg·mL^−1^ was found to be the minimum concentration not inducing germination inhibition (data not shown). After 24 h of soaking, the seeds were placed in a filter paper to reduce their water content.

### 4.5. Plant Growth Conditions

After being primed, the seeds were sown in trays using a substrate of sand:peat (1:1). Trays were transferred to a growth chamber at a temperature of 23/18 °C with 16 h of daylight and 8 h of night and 70% relative humidity under a light intensity of 120 μmol m^−2^ s^−1^ in a complete randomized block design. Seeds were irrigated with distilled water until emergence. Afterwards, plants were continuously irrigated with convenient nutrient solution (2, 4 or 8 dS·m^−1^) to maintain optimal substrate moisture until the end of the experiment. Plants were harvested when they had achieved the 4-leaves level.

#### 4.5.1. Determination of Plant Shoot and Root Biomass

Following the harvest of tomato seedlings, the fresh biomass and shoot length were measured and the root part was analyzed with the WhinRhizo (Regent Instruments) software.

#### 4.5.2. Salinity Stress Application

After seeds germination, plants were irrigated with a full strength Hoagland nutrient solution [52] with three different electrical conductivities (EC) (2, 4, 8 dS·m^−1^). EC was adjusted using NaCl.

#### 4.5.3. Determination of Antioxidant Activity

Plant extract was prepared by extracting 0.5 g of leave biomass in 5 mL of 80% methanol. The antioxidant activity was estimated using the 2,2-Diphenyl-1-picrylhydrazyl (DPPH) radical scavenging protocol developed by Blois [53]. Briefly, 750 µL of plant extract was added to an equal volume of 2 mM methanol DPPH solution. The mixture was stirred and incubated for 60 min at room temperature. The absorbance was determined at 517 nm. Ascorbic acid was used as a standard, and antioxidant activity was expressed as mg ascorbic acid g^−1^ FW.

#### 4.5.4. Hydrogen Peroxide (H_2_O_2_) Concentration

The hydrogen peroxide concentration was determined as described by Velikova et al. [54]. The amount of 100 mg of leaf biomass was homogenized with 1 mL 0.1% (*w/v*) trichloroacetic acid (TCA). The mixture was centrifuged at 12,000× *g* for 15 min at 0 °C and 500 µL of the supernatant was added to 500 µL 10 mM potassium phosphate buffer (pH 7.0) and 1 mL 1 M of potassium iodide. Absorbance was read at 390 nm. H_2_O_2_ was used as a standard and values were given as nmol·g^−1^ FW.

#### 4.5.5. Soluble Sugars and Total Proteins Content

Plant extracts were obtained by adding 60 mg of leaf biomass to 2 mL of water. Soluble sugars were estimated as described earlier. Proteins were quantified using the Bradford assay [55]. The amount of 0.1 mL of plant extract was added to 5 mL of Bradford reagent in a test tube. The homogenate was mixed, and the absorbance was read at 595 nm after 2 min. Bovine serum albumin (BSA) was used as a standard and proteins values were given as mg equivalent BSA·g^−1^ FW.

#### 4.5.6. Chlorophyll a, b and Total Carotenoids

Chlorophyll *a*, *b* and total carotenoids were determined according to Lichtenthaler [56]. The amount of 20 mg of leaf biomass was added to 5 mL of acetone 80% (*v*/*v*), fully grinded and centrifuged at 500× *g* for 5 min at room temperature. The supernatant was recovered and absorbance read at 662 nm, 642 nm, and 470 nm. Values were calculated using the following formulas:
Chlorophyll *a* (μg mL^−1^) = 12.5 *A*_663_ − 2.79 *A*_646_
Chlorophyll *b* (μg mL^−1^) = 21.5 *A*_646_ − 5.10 *A*_663_
$$\mathrm{Total}\;\mathrm{carotenoids}=\frac{1000A_{470}-1.82ch\left(a\right)-85.02ch\left(b\right)}{198}$$

### 4.6. Statistical Analyses

A completely randomized block design with two factors (salinity levels and extract fractions) and three replicates was adopted for this experiment. Data were analyzed using two-way ANOVA followed by Tukey’s multiple tests using the Minitab 19 statistical software. A principal component analysis (PCA) was done using the same software.

## 5. Conclusions

Priming tomato seeds with *Ulva lactuca* extract at a concentration of 1 mg·mL^−1^ succeeded in minimizing the deleterious effect of salinity stress on tomato seedlings by increasing their fresh weight in comparison to untreated plants. This amelioration was associated with different physiological alterations, such as decreasing ROS production, equilibrating plants antioxidant activity, enhancing total proteins and sugars, and improving the chlorophyll content. Solvent–solvent fractionation enabled us to establish a direct correlation between some compounds and the salinity stress alleviation, namely glycine betaine and phenolic compounds, suggesting that a target extraction methodology is possibly more adequate to obtain a desired effect on specific growth conditions (abiotic or biotic stress). This study demonstrated that the *Ulva lactuca* extract would be an effective PB for mitigating abiotic stresses menacing crops.

## Figures and Tables

**Figure 1 plants-10-01104-f001:**
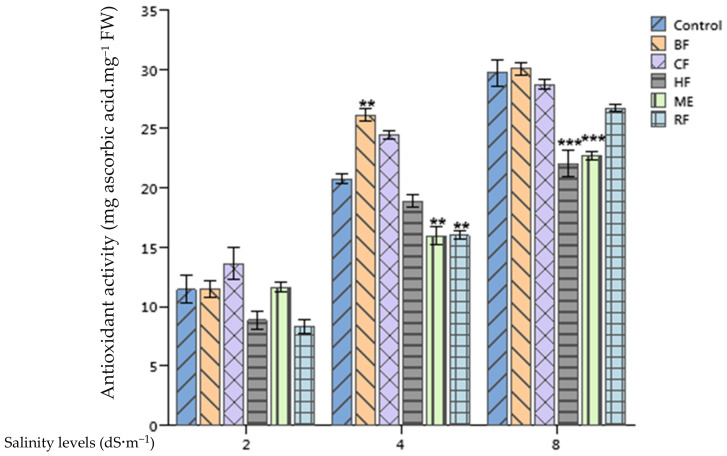
Effect of priming tomato seeds with *Ulva lactuca* methanol extract (ME), hexane fraction (HF), chloroform fraction (CF), n-butanol fraction (BF), and residual fraction (RF) under different salinity levels (2, 4, 8 dS·m^−1^) on the antioxidant activity of tomato leaves. Mean ± Standard deviation (SD) values followed by asterisks indicate significant differences between treatments and the control (** *p* < 0.01; *** *p* < 0.001).

**Figure 2 plants-10-01104-f002:**
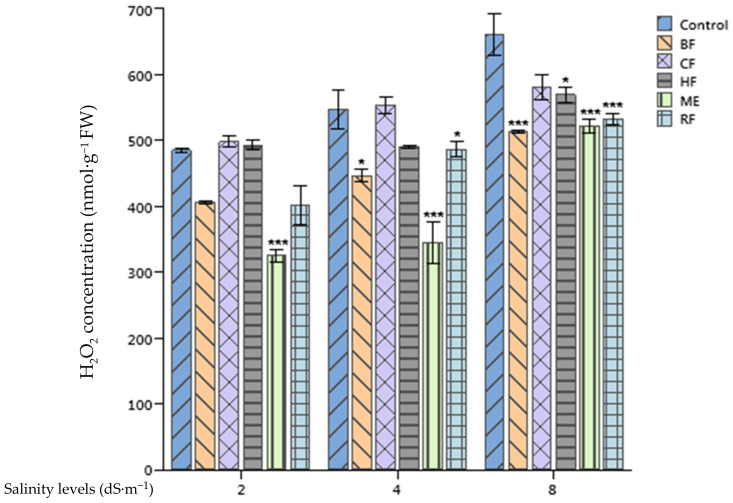
Effect of priming tomato seeds with *Ulva lactuca* methanol extract (ME), hexane fraction (HF), chloroform fraction (CF), n-butanol fraction (BF), and residual fraction (RF) under different salinity levels (2, 4, 8 dS·m^−1^) on hydrogen peroxide concentration of tomato leaves. Mean ± SD values followed by asterisks indicate significant differences between treatments and the control (* *p* < 0.05; *** *p* < 0.001).

**Figure 3 plants-10-01104-f003:**
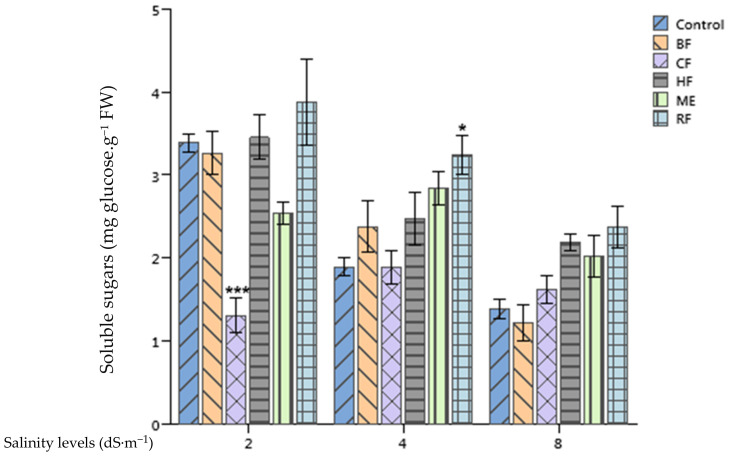
Effect of priming tomato seeds with *Ulva lactuca* methanol extract (ME), hexane fraction (HF), chloroform fraction (CF), n-butanol fraction (BF), and residual fraction (RF) under different salinity levels (2, 4, 8 dS·m^−1^) on soluble sugars content of tomato leaves. Mean ± SD values followed by asterisks indicate significant differences between treatments and the control (* *p* < 0.05; *** *p* < 0.001).

**Figure 4 plants-10-01104-f004:**
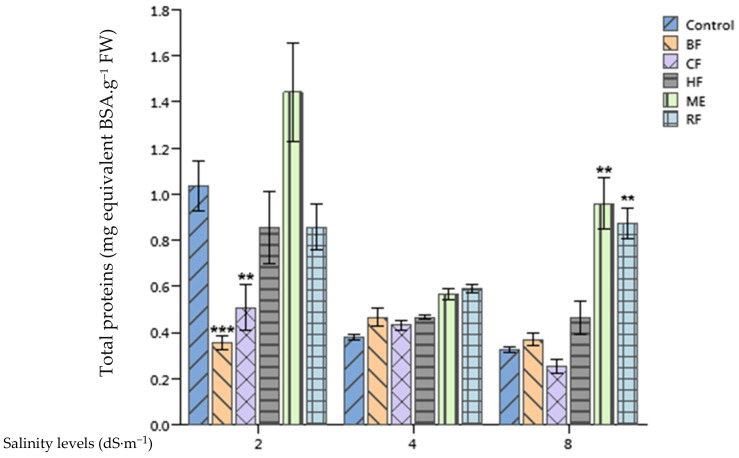
Effect of priming tomato seeds with *Ulva lactuca* methanol extract (ME), hexane fraction (HF), chloroform fraction (CF), n-butanol fraction (BF), and residual fraction (RF) under different salinity levels (2, 4, 8 dS·m^−1^) on total proteins content of tomato leaves. Mean ± SD values followed by asterisks indicate significant differences between treatments and the control (** *p* < 0.01; *** *p* < 0.001).

**Figure 5 plants-10-01104-f005:**
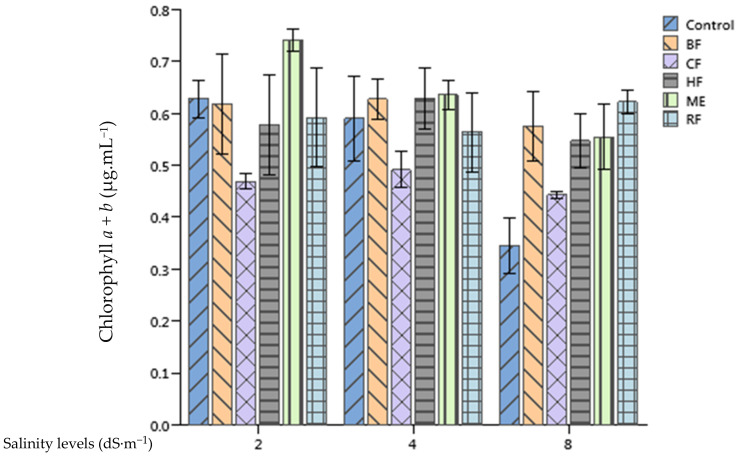
Effect of priming tomato seeds with *Ulva lactuca* methanol extract (ME), hexane fraction (HF), chloroform fraction (CF), n-butanol fraction (BF), and residual fraction (RF) under different salinity levels (2, 4, 8 dS·m^−1^) on total chlorophyll content of tomato leaves.

**Figure 6 plants-10-01104-f006:**
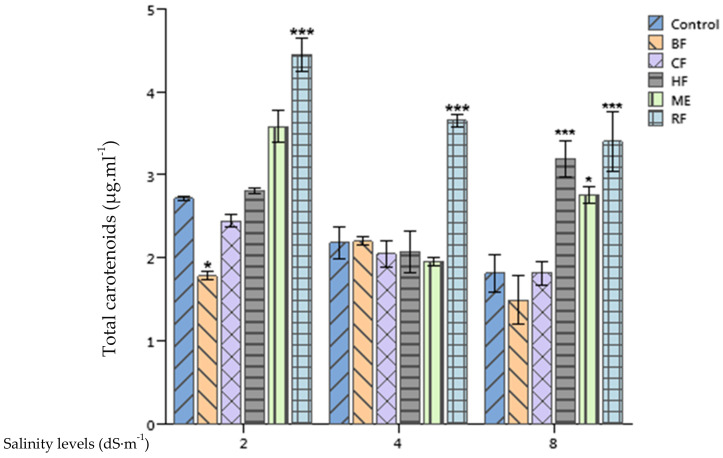
Effect of priming tomato seeds with *Ulva lactuca* methanol extract (ME), hexane fraction (HF), chloroform fraction (CF), n-butanol fraction (BF), and residual fraction (RF) under different salinity levels (2, 4, 8 dS·m^−1^) on total carotenoids concentration of tomato leaves. Mean ± SD values followed by asterisks indicate significant differences between treatments and the control (* *p* < 0.05; *** *p* < 0.001).

**Figure 7 plants-10-01104-f007:**
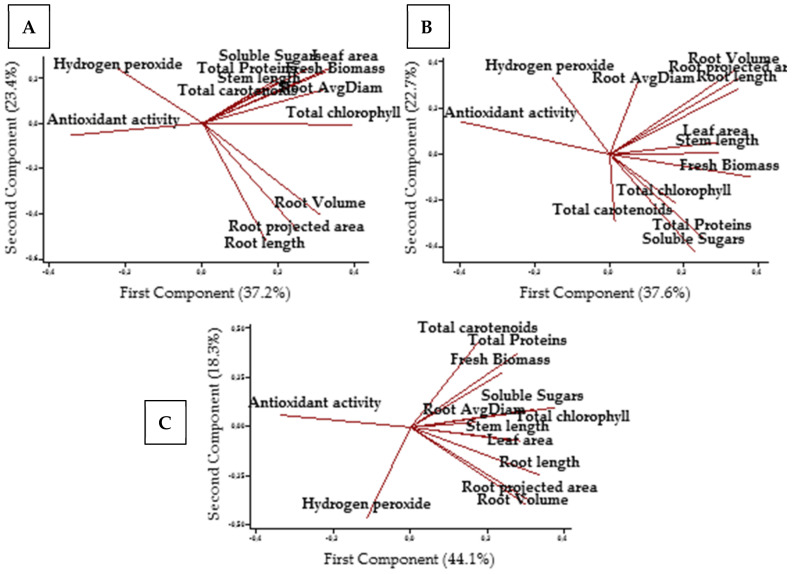
Principal component analysis of the antioxidant activity, hydrogen peroxide concentration, total chlorophyll, soluble sugars, total proteins, total carotenoids, leaf area, fresh biomass, root projected area, root length, root volume, root average diameter (Avgdiam) at 2 dS·m^−1^ (**A**), 4 dS·m^−1^ (**B**), and 8 dS·m^−1^ (**C**).

**Table 1 plants-10-01104-t001:** Biochemical characterization of *Ulva lactuca* methanol extract (ME), hexane fraction (HF), chloroform fraction (CF), n-butanol fraction (BF), and residual fraction (RF).

Extract Fraction	Total Phenols Content (mg GAE·g^−1^ DW)	Total Flavonoids (mg Quercitine·g^−1^ DW)	Glycine Betaine (mg·g^−1^ DW)	Soluble Sugars (mg Glucose·g^−1^ DW)	Extraction Yield (%)
ME	112.640 ^a^	4.870 ^a^	7.290 ^b^	0.296 ^a^	20.47 ^bc^
HF	84.047 ^b^	3.073 ^b^	1.97 ^e^	0.140 ^c^	24.46 ^b^
CF	54.057 ^c^	1.953 ^c^	3.79 ^c^	0.156 ^e^	5.99 ^d^
BF	61.904 ^c^	1.230 ^d^	2.293 ^d^	0.150 ^d^	18.65 ^c^
RF	79.971 ^b^	1.955 ^c^	8.033 ^a^	0.178 ^b^	49.39 ^a^

Groups with different letters are significantly different (*p* < 0.05).

**Table 2 plants-10-01104-t002:** Effect of *Ulva lactuca* methanol extract (ME), hexane fraction (HF), chloroform fraction (CF), n-butanol fraction (BF), and residual fraction (RF) application on growth traits of tomato plants growing in different salinity levels (2, 4, 8 dS·m^−1^). The control corresponds to plants with seeds primed in distilled water. Values are given as mean ± Standard deviation (SD).

**Fractions**	**Shoot Weight (g)**			**Leaf Area (cm^2^)**			**Root Length (cm)**		
	**2 dS·m^−1^**	**4 dS·m^−1^**	**8 dS·m^−1^**	**2 dS·m^−1^**	**4 dS·m^−1^**	**8 dS·m^−1^**	**2 dS·m^−1^**	**4 dS·m^−1^**	**8 dS·m^−1^**
Control	2.49 ± 0.31	1.68 ± 0.28	1.22 ± 0.15	94.86 ± 7.29	85.82 ± 5.27	73.18 ± 6.79	747.4 ± 99.8	756.3 ± 221.4	419.7 ± 22.7
ME	2.68 ± 0.44	2.35 ± 0.23 *	1.44 ± 0.21	100.38 ± 6.55	94.8 ± 28.8	79.4 ± 18	1021 ± 233	872.4 ± 167.9	509.4 ± 113.4
HF	2.94 ± 0.33	2.09 ± 0.19	1.42 ± 0.36	104.61 ± 8.76	93 ± 20.2	73.51 ± 6.2	828 ± 152.8	877.2 ± 72.5	628.4 ± 15.9
CF	2.21 ± 0.46	1.52 ± 0.19	1.04 ± 0.22	81.56 ± 9.56	72.26 ± 4.36	55.84 ± 5.91	917 ± 188	747.4 ± 99.8	419.7 ± 22.3
BF	1.78 ± 0.34 *	1.36 ± 0.22	1.18 ± 0.42	82.66 ± 8.58	68.44 ± 3.11	55.76 ± 8.71	939.4 ± 32.1	483.1 ± 96.8	450.5 ± 39.6
RF	3.10 ± 0.38	2.24 ± 0.37	2.08 ± 0.32 **	111.8 ± 21.1	83.47 ± 12.62	82.03 ± 10.35	1001.3 ± 38.6	756.3 ± 221.4	499.6 ± 15.6
**Fractions**	**Root Projected Area (cm^2^)**			**Root Volume (cm^3^)**			**Average Diameter (mm)**		
	**2 dS·m^−1^**	**4 dS·m^−1^**	**8 dS·m^−1^**	**2 dS·m^−1^**	**4 dS·m^−1^**	**8 dS·m^−1^**	**2 dS·m^−1^**	**4 dS·m^−1^**	**8 dS·m^−1^**
Control	18.6 ± 2.08	18.03 ± 5.01	10.73 ± 0.54	0.36 ± 0.02	0.34 ± 0.05	0.22 ± 0.01	0.25 ± 0.01	0.24 ± 0.01	0.25 ± 0.01
ME	25.41 ± 6.25	20.38 ± 3.88	12.47 ± 2.83	0.50 ± 0.08	0.37 ± 0.04	0.24 ± 0.03	0.25 ± 0.01	0.23 ± 0.01	0.24 ± 0.01
HF	19.6 ± 2.87	20.49 ± 1.58	15.79 ± 1.94	0.36 ± 0.03	0.38 ± 0.02	0.31 ± 0.04	0.24 ± 0.01	0.23 ± 0.01	0.25 ± 0.02
CF	21.05 ± 4.43	18.6 ± 2.08	10.714 ± 0.49	0.38 ± 0.05	0.36 ± 0.02	0.21 ± 0.01	0.23 ± 0.01	0.25 ± 0.01	0.26 ± 0.01
BF	22.29 ± 0.34	10.75 ± 2.18	10.70 ± 1.28	0.42 ± 0.01	0.19 ± 0.02	0.20 ± 0.02	0.24 ± 0.01	0.22 ± 0.01	0.24 ± 0.01
RF	25.18 ± 0.28	18.03 ± 5.01	11.56 ± 0.77	0.50 ± 0.01	0.34 ± 0.05	0.24 ± 0.02	0.25 ± 0.01	0.24 ± 0.01	0.28 ± 0.01

Means ± SD values followed by asterisks indicate significant differences between treatments and the control (* *p* < 0.05; ** *p* < 0.01).

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
