# Peer review of "Ulva lactuca Extract and Fractions as Seed Priming Agents Mitigate Salinity Stress in Tomato Seedlings"

_plants, 2021, doi:10.3390/plants10061104_

Round 1

Reviewer 1 Report

The manuscript deals with the very important issue of tomato seeds priming with the Ulva lactuca extract under salinity stress conditions. The authors sought to explain the mechanisms of action of algae stimulants on the plant. This is an undoubted advantage of this article, because most scientific works are limited to finding the effect of various stimulants on the plant, without investigating the mechanisms of their action.

The scope of research is wide and has been well thought out. The scope of research is extensive and well thought out. The results were presented correctly. My only remark is that minor clarifications were omitted (highlighted below). Discussion is written properly.

Remarks:

  1. No explanation of what it means: Extraction Yield in table 1
  2.  In table 2. there is no explanation what C means. I know that it is control, but in Methods there is no explanation of what constituted the control.
  3. There are errors in the numbering of the figures. Figure 1 is twice.
  4. No explanation of what constituted First component and Second component (Figure 7).
  5. Line 358: Bradford, 1975 should be cited as [53] and should be added to References
  6. Line 357 There are used twice word: analysis, one of them is unnecessary.  

Author Response

Reviewer comment

Action

Extraction yield explanation not present at table 1

Meaning was detailed in material and methods part, page 11, lines 315 and 316.

No explanation what C means

A description of the control was added at material and methods, page 12, line 351. C was changed to ‘control’ in all tables and figures.

Errors in the numbering of the figures

Errors were corrected.

No explanation of what constituted First component and Second component

Parameters that constituted First and second component were listed in page 8 lines 182-184

Line 358: Bradford, 1975 should be cited as [53] and should be added to References

Error corrected as requested.

Line 357 There are used twice word: analysis, one of them is unnecessary.

Error corrected.

Reviewer 2 Report

Comments and Suggestions for Authors

The Manuscript provided the investigation of the effect of Ulva lactuca extract as seed-priming agent for tomato plants under optimal and salinity stress conditions. The instrumentation is state-of-the-art and the article is suitable for Plants.

However, there are some flaws surrounding this piece of work.

While the text is relatively well written, there is inconsistency in the writing of units in the text, missing spaces between words, missing italics (in latin words, chlorophyl a, b), etc. The work requires careful stylistic control. In the manuscript it is necessary to unify the expressions "total polyphenols" and "total phenols".

In the figures I suggested to put in the first place control, then the priming variants. I suggest polishing the figures and removing the black frame., especially in Figures 6

It is not entirely clear from the description of the methodology how the stress conditions were applied. It was a single dose of modified Hoagland's solution, or it was a repeated application. When exactly were the solutions applied.

It is not clear how the individual measured data were expressed, for example for total phenols the expression is in mg of gallic acid equivalents per g ... please explain per gram of what, fresh weight, dry weight ?

Page 12, line 319 - include citation in bibliography
page 13, line 358 - include citation in bibliography

Author Response

Reviewer comment

Action

Inconsistency in the writing of units

Units were homogenized

Missing italics and spaces

Missing italics and spaces were corrected.

Figures format

Figures were adapted to the suggested changes.

Salinity treatments application frequencies

Salinity treatments were applied continuously from emergence to harvesting tomato plants. A description was added to material and methods, page 12 lines 357-360.

Problem of Dry or fresh weight in expressions

Dry or fresh weight expressions were added.

Page 12, line 319 - include citation in bibliography
page 13, line 358 - include citation in bibliography

Citations were added in bibliography.

Reviewer 3 Report

Overall, the paper is clearly written and the described experiments have been properly designed. Careful attention has been given to perform the statistical analysis of the collected data. I think this is an interesting contribution to the domain of seed priming with SEs. However, the Methods section should be completed because some important details are missing. The Disscussion should also be completed, particularly adding a paragraph about the possible molecular mechanism underlying glycine betaine’s action. I herafter offer a list of suggestions that I hope will help the authors for the next version of the manuscript.

- You are interested in the effects of priming tomato seeds with SEs. Why did you decide to apply the salt stress after germination and not before of (or during) germination, i.e. couldn’t you be missing some benefits of the priming treatment on e.g. germination rates facing salt stress or seedlings early survival facing salt stress? And on the other extreme : the main agronomic trait of interest for tomatoes is fruits production. Why did you decide to harvest plantlets at the four leaves stage (line 335) and how can you be sure that the detected benefits are actually agronomically relevant and could translate into an increased fruit fitness ? Please add a couple of sentences in Discussion to explain your choice.                 

- Table 2 (lines 98-104) : The only statistically significant improvement (based on two-way ANOVAs, n=3) for the ME compared to the control extract is seen considering shoots weight at 4 ds.m-1. Do you have any idea why a similar statistically significant improvement is not seen at 8 dS.m-1, i.e. do you think there is a biological reason behind that or it could simply be due to the low number of biological replicates used here? And same question for the RF extract, which only yields a significant increase in shoots weight at 8 dS.m-1. Overall, p values are a useful indicator but have to be taken with care particularly when using such a low number of biological replicates.

- Missing in discussion : a few sentences explaining what could be the mechanism of action of glycine betaine at the molecular level. I know there are excellent reviews about the topic but it would be helpful to give a little reminder here.

Here are some more specific comments :

Table 2 : How was the control made exactly, were these tomato seeds that were primed in water only ? I haven’t found the information on Materials and Methods.

Line 301 : the differents fractions had previously been dried (line 296). How were the samples prepared here, i.e. which solvant was used to resuspend the dried factions and which volume was employed ?

Line 303 : How was the standard curve made, i.e. did you prepare a dilution series of gallic acid ?

Lines 305-310 : « 1 ml of plant extract » same comment as for line 301, how were the dried seaweed extracts resuspended ? How was the standard curve made ?

Line 327 : How was the priming performed here, i.e. were the seeds incubated in an aqueous solution containing 1mg of dried extract per ml ? Because some of the extracts are expected to contain highly hydrophobic molecules, how where these molecules maintained in solution ?

Line 344 : How was the « plant extract » prepared, i.e. which quantity of leaves was required, how were the leaves broken and which solvant was employed ? How was the standard curve prepared ?

Table I (line 94): SD values are missing from the table.

Figure 1 (line 132): This is not Figure 1 and Figure 2. All figure numbers have to be changed accordingly.

Author Response

Reviewer comment

Action

Why salinity treatments were not applied starting before germination? Why experiment was not maintained until fruit harvest?

Most of studies dealing with seed priming were mainly focused on germination stage. Moreover, the effect of SE on plants would be different from a stage to another. Consequently, our study focused on the seedling tomato stage to understand to some extent the effect of Ulva lactuca extract as a priming agent on tomato early vegetative growth. 

Fresh weight improvement significant at 4ds.m-1 and not at 8 dS.m-1 for the ME. While the opposite was recorded for the RF.

Seaweed extract biostimulants are thought to enhance yield or/and biomass mainly in stress conditions. Absence of significance in one of the 2 salinity levels (4 or 8 dS.m-1) would be related to the size of the standard deviation.

Glycine betaine at molecular level

A paragraph was added to discussion.

Control description

A description of the control was added at material and methods, lines 327 and 328. C was changed to ‘control’ in all tables and figures.

Which solvent was used to resuspend the dried factions and which volume was employed?

Extracts were resuspended in water in a ratio of 1 mg dry extract / 1 ml water. No other solution was added to prepare the extracts. Details were added to material and methods, page 12, lines 352 and 353.

How was the standard curve made, i.e. did you prepare a dilution series of gallic acid?

Standard curve was prepared using dilution series from a Gallic acid stock solution (1 mg.ml-1).

Standard curves for other parameters

Descriptions were added to material and methods for each parameter.

Missing SD values from table 2

SD values were lesser than 0.005. Consequently, when fixing the number of decimals to 2, it gives 0.00 as a SD. That is why we removed them.

Figures numbers have to be changed

Error was corrected.